# Tumor BRCA Testing in Epithelial Ovarian Cancers: Past and Future—Five-Years’ Single-Institution Experience of 762 Consecutive Patients

**DOI:** 10.3390/cancers14071638

**Published:** 2022-03-23

**Authors:** Caterina Fumagalli, Ilaria Betella, Alessandra Rappa, Maria di Giminiani, Michela Gaiano, Luigi Antonio De Vitis, Benedetta Zambetti, Davide Vacirca, Francesco Multinu, Konstantinos Venetis, Nicoletta Colombo, Massimo Barberis, Elena Guerini Rocco

**Affiliations:** 1Clinical Unit of Oncogenomics, European Institute of Oncology (IEO), Istituto di Ricovero e Cura a Carattere Scientifico (IRCCS), 20141 Milan, Italy; caterina.fumagalli@asst-valleolona.it (C.F.); alessandra.rappa@ieo.it (A.R.); davide.vacirca@ieo.it (D.V.); elena.guerinirocco@ieo.it (E.G.R.); 2Department of Diagnostic Services, Division of Pathology, Azienda Socio Sanitaria Territoriale (ASST) Valle Olona, 21013 Gallarate, Italy; 3Division of Gynecologic Surgery, European Institute of Oncology (IEO), Istituto di Ricovero e Cura a Carattere Scientifico (IRCCS), 20141 Milan, Italy; ilaria.betella@ieo.it (I.B.); maria.digiminiani@unimi.it (M.d.G.); michela.gaiano@unipr.it (M.G.); luigiantonio.devitis@ieo.it (L.A.D.V.); benedetta.zambetti@ieo.it (B.Z.); francesco.multinu@ieo.it (F.M.); 4Department of Health Sciences, University of Milan, 20122 Milan, Italy; 5Obstetrics and Gynaecology Unit, Department of Medicine and Surgery, University of Parma, 43126 Parma, Italy; 6Obstetrics and Gynaecology Unit, Department of Medicine and Surgery, University of Milano Bicocca, 20900 Monza, Italy; 7Department of Oncology and Hemato-Oncology, University of Milan, 20122 Milan, Italy; konstantinos.venetis@unimi.it; 8Division of Pathology, European Institute of Oncology (IEO), Istituto di Ricovero e Cura a Carattere Scientifico (IRCCS), 20141 Milan, Italy; 9Division of Gynecologic Oncology, European Institute of Oncology (IEO), Istituto di Ricovero e Cura a Carattere Scientifico (IRCCS), 20141 Milan, Italy; nicoletta.colombo@ieo.it; 10School of Medicine and Surgery, Università degli Studi di Milano-Bicocca, 20126 Milan, Italy

**Keywords:** BRCA, epithelial ovarian carcinoma, NGS

## Abstract

**Simple Summary:**

Tumor BRCA testing is crucial in the clinical management of women affected by epithelial ovarian cancer (EOC). In the present study, we aimed to report the results of five years of experience in tumor BRCA testing performed in a single-institution diagnostic setting. We profiled 762 consecutive EOC patients with a failure rate of less than 1% and less than two weeks of turnaround time, which is consistent with the clinical needs. We identified 23.4% of cases with pathogenic/likely pathogenic mutations, including 76% of patients affected by germline and 24% by somatic alterations. Here, we proposed a comprehensive and multidisciplinary clinical workflow that could be successfully followed for the identification of somatic as well as germline alterations, maximizing the benefit of BRCA testing both from a therapeutic and risk assessment perspective.

**Abstract:**

The establishment of PARP inhibitors in the treatment of epithelial ovarian carcinoma (EOC) has prompt BRCA assessment at the time of diagnosis. We described our five years of experience of tumor BRCA testing, as part of a multidisciplinary workflow for the management of EOC patients. We used a BRCA next-generation sequencing (NGS) test for profiling formalin-fixed, paraffin-embedded (FFPE) EOCs of 762 consecutive patients, with a success rate of 99.7% and a median turnaround time of 12 days. We found 178 (23.4%) cases with pathogenic/likely pathogenic (P/LP) mutations, 74 (9.7%) cases with variants of uncertain significance and 508 (66.8%) wild type tumors. Among 174 patients without P/LP mutations and investigated with multiple-ligation probe-amplification analysis on peripheral blood, two (1.1%) were positive for large rearrangements. Patients with P/LP alterations and/or with positive family history were referred to genetic counselling. Comparing tumor and blood NGS test results of 256 patients, we obtained a tumor test negative predictive value of 100% and we defined 76% of P/LP alterations as germline and 24% as somatic variants. The proposed workflow may successfully identify EOC patients with BRCA1/2 alteration, guiding both therapeutic and risk assessment clinical decisions.

## 1. Introduction

Precision medicine aims to personalize treatments despite the “drug and disease dyad” approach. Precision oncology encompasses tumor molecular alterations, genomic background, and immune profiles along with patient’s clinicopathological characteristics, to tailor the most effective treatment for each patient. In the last few years, tumor genetic profiling along with the introduction of poly ADP-ribose polymerase (PARP) inhibitors in clinical practice has moved a step forward to precision oncology for women affected by epithelial ovarian carcinoma (EOC) [1,2,3]. On the one side, the detection of BRCA1/2 alterations allows for identifying women with better prognosis [4], who may benefit from platinum-based therapy and with a higher probability of PARP inhibitors’ response [5,6,7,8]. On the other side, maintenance therapy with PARP inhibitors has revolutionized the treatment of newly diagnosed and recurrent disease, achieving an unprecedented improvement in patient outcomes [5,6,7,8,9,10]. Thus, different scientific Societies, including the American Society of Clinical Oncology (ASCO), the National Cancer Comprehensive Network (NCCN), the Society of Gynecologic Oncology (SGO), the European Society of Medical Oncology (ESMO) and the AssociazioneItaliana di Oncologia Medica (AIOM) have recommended BRCA1/2 testing at the time of EOC diagnosis [11,12,13,14]. Guidelines for Next Generation Sequencing (NGS)-based BRCA test implementation have been proposed [15,16,17,18] and the feasibility of tumor BRCA assessment has been recently documented [19,20,21,22,23]. However, some critical issues still need to be addressed. Despite the improvements in NGS data analysis algorithms, the detection of large rearrangements (LRs) in formalin-fixed paraffin-embedded (FFPE) specimens remains sub-optimal. Moreover, the assignment of variant clinical significance is challenging, especially for variants with conflicting interpretations or without pathogenicity data reported that may consequently increase the quota of VUS (variants with unknown clinical significance). Finally, the clinical management of women affected by EOC should be designed to offer the most effective care for every single patient, with a comprehensive and multidisciplinary approach. Therefore, it should include genetic counselling and the ascertainment of germline or somatic nature of every BRCA1/2 alteration identified in the tumor to direct the risk assessment and the following preventive actions for the woman itself and her relatives.

Previously, we reported the feasibility of implementing a tumor NGS-based BRCA test in the clinical setting [20]. In the present study, we described the five years of single-institution experience of tumor BRCA testing, including a cohort of 762 consecutive women affected by EOCs. Thus, we sought (i) to report the prevalence of BRCA1/2 variants according to the pathogenicity level and in association with clinicopathological features; (ii) to evaluate the prevalence and the characteristics of somatic and germline alterations comparing tumor and blood BRCA test results; and (iii) to describe the clinically driven workflow for the management of EOC patients.

## 2. Materials and Methods

### 2.1. Study Population

This single-institution, retrospective study included a total of 762 women with a diagnosis of non-mucinous and non-borderline EOC who were referred to the Clinical Unit of Oncogenomics of the European Institute of Oncology (IEO) from October 2016 to June 2021 for tumor BRCA1/2 analysis. Each patient gave written informed consent, which obtained Institutional Review Board approval (GNM.MO.3003.A, last revised version 9 December 2019).

The clinicopathological characteristics were abstracted from the electronic clinical records. Demographic data included age at surgery, clinical data included grade, stage and family history of cancer. Stages were assigned according with FIGO 2014 [24] and additionally categorized as early stage (stages I or II A) and advanced (stages IIB or III or IV) [13]. The clinicopathological characteristics were reported in Table 1. The family history was considered positive when at least 1 first-degree relative had a BRCA1/2-related cancer diagnosis [20].

### 2.2. Next Generation Sequencing-Based BRCA Tumor Testing

BRCA tumor testing was performed using the NGS panel “Oncomine BRCA Research Assay” (ThermoFisher, Waltham, MA, USA), as previously reported [20]. Briefly, 10 ng of DNA extracted from representative formalin-fixed paraffin-embedded (FFPE) tumor tissue blocks were used for library preparation and the subsequent chip loading was performed automatically on the Ion Chef System (ThermoFisher, Waltham, MA, USA). The sequencing run was done in duplicate using Ion S5 System (ThermoFisher, Waltham, MA, USA). The data analysis was carried out with the Ion Reporter Analysis Software. Variants with an allele frequency ≥5% and with adequate quality metrics were visually inspected using the Integrative Genomics Viewer (IGV) software. The variants were classified according to the five clinical class system proposed by IARC (International Agency for Research on Cancer). The BRCA Exchange database was consulted, and the variant clinical significance was assigned according to the ENIGMA consortium revision [25,26]. If the variant was reported as “not yet reviewed” by the ENIGMA consortium, other databases were questioned as ClinVar [27] or Leiden Open source Variation Database (LOVD) [28] combined with Functional Assay Results and In Silico Prior Probability of Pathogenicity obtained with computational algorithms.

### 2.3. Multiplex Ligation-Dependent Probe Amplification (MLPA)

MLPA analysis was performed on peripheral blood, aiming to detect BRCA1/BRCA2 large rearrangements. The SALSA MLPA probemix P002 and SALSA MLPA probemix P009 (MRC Holland, Amsterdam, The Netherlands) kits were used for the evaluation of BRCA1 and BRCA2, respectively. The analysis was performed on the 3500Dx Genetic Analyzer (Thermo Fisher Scientific, Waltham, MA, USA) and the Coffalyser.Net tool (MRC Holland, Amsterdam, the Netherlands) was used for data analysis. The BRCA1 exon numbering used in this P002 BRCA1 probemix was the traditional exon numbering (exons 1a, 1b, 2, 3, and 5–24), wherein no exon 4 is present.

### 2.4. Statistical Analysis

Statistical analysis was carried out using SPSS Statistic 25 software. Chi-Square test with Yates correction and *t*-test calculators were used for data comparison of categorical variables. *p*-values < 0.05 were considered statistically significant.

## 3. Results

### 3.1. Tumor BRCA Testing Results

All the 762 FFPE tumor samples were considered adequate for the NGS test, according to the tumor cell content (more than 10%) and the DNA yield (more than 10 ng). Two cases (0.3%) failed the NGS analysis since they did not meet the NGS run quality parameters. Overall, the median turnaround time (TAT) was 12 calendar days (range 4–50 calendar days) from test request to final molecular report.

Among the 760 tumor samples successfully sequenced, 178 (23.4%) cases harbored a pathogenic/likely pathogenic (P/LP) mutation, 74 (9.7%) cases showed a variant of uncertain significance (VUS) only, and 508 (66.8%) were defined as BRCA1 and BRCA2 wild type (wt) (Figure 1). In detail, 123 (69%) tumors had P/LP alteration affecting BRCA1, including one sample with two concurrent pathogenic mutations in BRCA1, whereas 55 (31%) cases showed a P/LP alteration in BRCA2. Overall, 179 P/LP mutations were identified. The genetic alterations spanned over the whole coding sequence with the overwhelming majority of them being concentrated in the exon 11 of both genes. In detail, 59 of 124 (47.6%) mutations of BRCA1 and 27 of 55 (49.1%) mutations of BRCA2 affected the exons 11 (Figure 2). Concurrent pathogenic mutations and VUS were observed in 8 cases, including 6 cases with both BRCA1 pathogenic variants and BRCA2 VUS and 2 cases with both BRCA2 pathogenic variants and BRCA1 VUS. Among the 74 tumor samples harboring VUS only, 36 (48.6%) cases had alterations in BRCA1, 37 (50%) tumors had VUS in BRCA2 and one case (1.4%) presented VUS in both BRCA1 and BRCA2 genes (Figure 1).

Comparing P/LP alterations and VUS, we observed a statistically significant higher variant allele frequency (VAF) in P/LP alterations (median VAF = 74%, range 5–99%) than in VUS (median VAF = 47%, range 5–98%) (*p* value < 0.00001) (Table 2). In particular, P/LP mutations affecting BRCA1 had a higher VAF compared to BRCA2 alterations (median BRCA1 VAF = 76.5%, median BRCA2 VAF = 63%), even though it is not statistically significant. Moreover, the prevalence of variant types was statistically different between P/LP alterations and VUS. The large majority of P/LP alterations were truncating mutations including frameshift or nonsense mutations (*n* = 147, 82.1% BRCA1 *n* = 101, BRCA2 *n* = 46) compared to missense mutations (*n* = 18, 10.1% BRCA1 *n* = 15, BRCA2 *n* = 3), alterations affecting the splice-sites (*n* = 7, 3.9% BRCA1 *n* = 2, BRCA2 *n* = 5), mutation spanning the intronic regions (*n* = 3, 1.7% BRCA1 *n* = 2, BRCA2 *n* = 1) or inframe deletions (*n* = 4, 2.2% BRCA1 *n* = 4). VUS were predominantly missense mutations (*n* = 37, 49.3%, BRCA1 *n* = 13, BRCA2 *n* = 24) or intronic alterations (*n* = 28, 37.3%, BRCA1 *n* = 17, BRCA2 *n* = 11) (Table 2).

### 3.2. Tumor BRCA1/2 Status According to Clinicopathological Characteristics

The tumor BRCA1/2 status according to clinicopathological parameters was reported in Table 3. The large majority of P/LP alterations occurred in high-grade serous carcinoma (94.4%), advanced tumors (85.4%), patients with FIGO stage III (64.6%) and family history positive (66.3%). Comparing P/LP, VUS, and wild type BRCA distribution, two clinicopathological features emerged as significantly associated (*p*-value < 0.05) with the presence of BRCA1/2 pathogenic alterations, namely the histological type and the family history. The prevalence of P/LP alterations settled at 23.4%, but the percentage increased to 26.5% in high-grade serous carcinoma and reached 27.5% in cases with familiar or personal history positive for BRCA-related cancer. However, 10 no-high grade serous carcinoma and 49 cases with negative familiar history harbored BRCA1/2 pathogenic alterations. No differences were observed regarding the age at the time of diagnosis of EOC among the cases with P/LP alteration (median 59, range 37–81) or cases harboring VUS (median 59, range 39–79) or wild type (median 62, range 24–84).

### 3.3. The “New Workflow” of Tumor BRCA Testing

We previously proposed a workflow for tumor BRCA testing in the clinical management of women affected by ovarian cancer [20]. Starting from January 2020, we implemented an institutional modified workflow, addressing the pending issue of large rearrangement (LR) detection. As shown in Figure 3, BRCA tumor testing was indicated for all women with a diagnosis of non-mucinous and non-borderline EOC, with results expected within four weeks. Patients with high grade serous and endometrioid EOC and positive BRCA tumor testing, including pathogenic/likely pathogenic variants only, were eligible for PARPi therapy and were additionally referred to genetic counseling. Patients with a negative tumor test or VUS were further investigated with a complementary test seeking for large rearrangements only. For these patients, a multiple-ligation probe-amplification (MLPA) analysis performed on peripheral blood was proposed. Applying the integrated workflow for 174 patients with negative tumor test, we identified 2 (1.1%) cases positive for large rearrangements. In detail, one case harbored a deletion in the BRCA1 gene, exon 21 and 22, the other case a deletion in BRCA1gene, exons 13–23. Moreover, 124 of 309 (40%) patients with negative tumor BRCA status and positive family history were referred to genetic counseling.

### 3.4. Comparison between Tumor and Blood BRCA Test Results

According to the described workflow, women with tumor BRCA positive results were addressed to genetic counseling. Among 178 tumor BRCA positive cases, 132 (74%) patients underwent blood BRCA testing and completed the genetic counseling. The remaining 46 (26%) cases included seven external patients referred to our institution for BRCA tumor testing only and nine patients who were undergoing genetic counseling. For 30 patients, the genetic counseling has been recommended but not yet performed. In addition, 124 patients with positive family history for BRCA-related cancer underwent germline BRCA testing even though they tested negative for tumor BRCA alterations (108 wild-type EOC and 16 VUS EOC). Overall, both tumor and blood test analyses were available for 256 patients. The results were reported in Table 4. The negative predictive value of BRCA tumor testing was 100%, as all the negative tumor BRCA results (BRCA wild type status) were confirmed in blood analysis. Moreover, tumor testing was able to detect additional BRCA variants in 42 cases, including 32 P/LP variants and 10 VUS. Furthermore, comparing the BRCA tumor testing and BRCA blood testing results, we discriminated somatic variants, detected in tumor specimens only, and constitutive-germline alterations confirmed in the blood sample. In our cohort 100 of 132 (76%), P/LP alterations were germline, whereas 32 of 132 (24%) were somatic variants. In detail, we observed a higher prevalence of somatic mutations affecting BRCA2 (*n* = 14 of 43, 33%) than BRCA1 (*n* = 18 of 89, 20%), even if not statistically significant (*p*-value = 0.12). Actually, a statistically significant higher Variant Allele Frequency (VAF) was observed for BRCA1/2 germinal alterations (median VAF = 77%, range 7–97%) compared to somatic mutations (median VAF = 38%, range 5–88%) (*p*-value < 0.00001) (Appendix A).

In Table 5, we reported the distribution of clinicopathological parameters according to the presence of somatic or germline pathogenic variants. Women affected by EOC and carriers of BRCA germline variants were younger (median age at onset 56 years) than patients with BRCA somatic alterations (median age at onset 62.5 years) (*p*-value = 0.01). Positive family history for BRCA-related cancer, FIGO stage III and the histological subtype (high-grade serous carcinoma) were significantly associated with the presence of a germline BRCA pathogenic alteration (*p*-value < 0.05). However, BRCA germline alterations occurred in 19 cases with no family history and in two patients affected by endometrioid carcinoma. Somatic alterations were equally distributed between cases with negative (46.9%) and positive (53.1%) family history, and 4 of 6 (66.7%) pathogenic variants found in endometrioid carcinoma were somatic mutations.

## 4. Discussion

The implementation of tumor BRCA testing in the clinical management of women affected by ovarian cancer has emerged as a fundamental clinical need. A robust and reliable tumor BRCA test allows for identifying both somatic and germline alterations, enlarging the identification of patients who most benefit from PARPi treatment. Scientific Societies supported the introduction of tumor BRCA testing in the routine diagnostic settings but few studies provided large data collected outside of clinical trials. After the implementation of tumor BRCA testing in our Institution [21], here we reported the result of five years of experience including 762 consecutive EOC patients.

We confirmed the feasibility of tumor BRCA testing in the routine clinical setting, with a very low failure rate (less than 1%) and a turnaround time consistent with clinical need (median TAT less than two weeks), strengthening our previous results [21] and in line with the recently reported experience of other Italian groups [19,23,29]. However, both the studies of Marchetti [23] and Turchiano [29] were based on a fresh-frozen tissue approach, which was a very promising strategy to overcome the degradation and chemical modification of DNA resulting from formalin fixation and paraffin embedding, but it required framework and facilities (i.e., biobanking) not always available in hospitals. In the present study, according to the results presented by the Ligurian BRCA working group [19], we reported a workflow based on the analysis of FFPE specimens, available in every Pathology Division. In this context, the robustness of tumor BRCA testing is crucial. In a study within the Italian NGS network, we recently demonstrated that BRCA tumor testing performed with different technologies in different centers may achieve the quality standards for reliability and reproducibility required in the clinical setting [30]. For 256 cases, the germline BRCA status was available. These cases consisted of 132 BRCA mutated tumors and 124 tumors without P/LP mutations. In this population, the concordance between tumor and blood tests is 100%, suggesting that the tumor BRCA test met the standards for clinical needs accomplishment. Recently, Hodgson et al. [31] and Callen et al. [32] reported a high concordance rate between tumor and germline BRCA testing within SOLO2 and PAOLA/ENGOT-ov25 phase III trials, supporting a wider implementation of tumor testing in ovarian cancer patients.

In the studied cohort, we observed P/LP mutations in 23% of tumors, mainly affecting the BRCA1 gene (69% BRCA1 vs. 31% BRCA2). The prevalence of P/LP alterations may vary in different populations or according to the geographical areas of origin of the analyzed patient’s cohort, from 6% of the Danish [33] to the 41% of the Ashkenazi Jewish [34]. Even among Italy, some specific areas, such as the Apulia region, were characterized by a higher incidence of BRCA alterations, reaching 39% [35]. In our referral Institute, we treated women with ovarian cancers coming from all the Italian regions and from other European or North-African countries and our prevalence of P/LP alterations were consistent with previous studies reporting a frequency of P/LP alterations from 19% to 32% in ovarian cancers, mainly of the BRCA1 (75–61%) gene [20,23,36,37].

In our series, the distribution of P/LP mutations spanned the whole coding sequence of BRCA1 and BRCA2 genes, but the large majority of P/LP alterations involved the exons 11 of both genes. The exons 11 were the largest in both BRCA1 and BRCA2, covering more than half of the gene coding sequence and containing ovarian cancer cluster regions [38,39,40]. The location of a pathogenic mutation may be helpful in the counseling of BRCA1/2 mutation carriers both for risk assessment and clinical decision-making (i.e., timing for risk-reducing salpingo-oophorectomy) [41] but also in predicting therapy response. Indeed, depending on the exon/domain affected, various isoforms may be produced that could retain a partial function or may be degraded via a nonsense-mediated decay (NMD) RNA pathway, leading to a different response to anti-cancer drugs. Even if further data are warranted [42], growing evidence suggests that BRCA1 exon 11 alterations may be associated with partial response or linked to therapeutic resistance to platinum-based and PARPi treatments [43,44].

The functional and clinical interpretations of variants of uncertain significance (VUS) pose a challenge for precision oncology. In our study, VUS alterations count for 10%. To further interpret VUS as P/LP or benign alterations, different approaches may be performed. The more reliable method may be the evaluation of the impact of a specific mutation via functional analyses (i.e., minigene construction) [45,46]; however, the classification of all possible variants by functional assays is laborious and time-consuming. Sequence- and structure-based computational tools (i.e., SIFT, PolyPhen2, PANTHER, FATHMMand I-Mutant), in particular the functionally validated sequence-based prediction models, may help assess the impact of numerous variants on protein activity [47,48]. A recent study led by Dines and colleagues [49] proposed to introduce in variant classification guidelines the “coldspot evidence” as well as “hotspots/critical domains” were considered evidence to support pathogenicity (ACMG/AMP PM1) [50]. The “coldspot” regions in BRCA1 and BRCA2 were defined as regions more tolerant to variation and thus the location of a missense variant within a coldspot region sustained its classification as likely benign rather than VUS. In our experience, whenever possible, the diagnostic report including a VUS result was accompanied with a comment regarding the pathogenicity level predicted by computational algorithms along with a description of the presence/absence of pathogenic or benign variants affecting the same or flanked codons.

According to the national guidelines [17], our institutional workflow proposed a BRCA tumor test to all patients at diagnosis of non-mucinous and non-borderline ovarian cancer, regardless of cancer family history. The test was suggested by skilled specialists who explained the value and purposes of tumor testing to the patients and obtained written informed consent. Even if the large majority of P/LP alterations was associated with high-grade serous carcinoma and patients with positive family history for BRCA-related cancer, limiting BRCA testing to these cases was an ineffective approach for identifying all patients eligible for PARPi, including patients with a germline pathogenic alteration, as recently reported in the meta-analysis conducted by Witjes and colleagues [51]. Indeed, we identified pathogenic variants in six endometrioid carcinoma, including two with germline alterations. We also observed 34 cases with P/LP variants and negative family history for BRCA-related cancer, including 19 patients harboring a germline alteration. The prevalence of BRCA somatic alterations considerably vary from 3–6% to 39% according to patient selection and clinico-pathological features [20,21,22,36,37,52] and in our series accounted for 24% of BRCA positive (P/LP variants) cases and 6% of the total population.

Starting from January 2020, the evaluation of BRCA large rearrangements on blood sample was offered for patients with VUS/negative tumor BRCA status. This additional step in the institutional workflow was introduced to overcome the limitations associated with amplicon-based technology and FFPE sample analysis in the CNV detection. Even if the large rearrangements accounted for a small quota of BRCA pathogenic variants (<10%), we found two positive cases among the 174 screened patients (1.1%), who may be eligible for PARPi treatment and referred to genetic counseling. During the genetic counselling, the geneticist may require a blood-based testing, describing to the patients the value of a blood test results in terms of risk assessment and following preventive actions for the woman itself and her relatives. For the blood-based BRCA testing, a specific written consent may be obtained.

Precision oncology has been expanding beyond BRCA1/2 testing and the evaluation of homologous recombination deficiency (HRD) as a biomarker associated with PARP inhibitors response represents an important clinical issue [53,54,55]. Indeed, although EOC that respond to platinum-based chemotherapy are eligible for maintenance regimens with PARP inhibitors, the benefit of this therapy varies among patients. Tumors characterized by HRD, mainly with loss-of-function mutations of *BRCA1* or *BRCA2*, benefit the most, while cases without HRD only moderately benefit.

To date, methods for HRD detection are limited to genetic profiling of homologous recombination repair genes or genomic assays evaluating HRD-induced genomic scar, even if emerging evidence supports the evaluation of RAD51 foci formation [56,57,58]. As recently reported, an NGS profiling including HR genes (*BARD1*, *BRIP1*, *PALB2*, *RAD51C*, and *RAD51D*) could improve the rate of tumor with HRD by 5–6%, identifying patients who may mostly benefit from PARPi therapy [29]. In our cohort, a comprehensive tumor profile including homologous recombination genes (i.e., *ATM*, *PALB2*, *RAD50*, *RAD51*, *RAD51B*, *RAD51C*, *RAD51D*, …) was performed for 100 cases, finding pathogenic alterations in four tumor samples, as previously reported [59]. However, HRD can be present even in the absence of such mutations. A specific DNA-aberration profile, defined as a genomic scar, including loss of heterozygosity (LOH), telomeric allelic imbalance (TAI) and large scale state transitions (LST), is caused by HRD. The genomic instability has been measured with NGS technology and FDA approved a specific diagnostic test, the Myriad myChoice HRD, that has been used in different clinical trials. In these months, many academic centers and biotech factories have been involved in design and manufacturing alternative assays that could be introduced in clinical practice. To date, commercial assays applicable in diagnostic laboratories have been placed on the market, with the potential to screen HRR genes along with genomic scar (i.e., Oncomine Comprehensive Assay Plus, ThermoFisher Scientific or HRD Focus Panel, Amoy Dx) even if a clinical validation is needed. In our diagnostic algorithm, we will improve the HRD evaluation beyond *BRCA1* and *BRCA2* and further analyses of HRD testing are currently ongoing.

In this view, in the near future, we will additionally change our clinical workflow, offering simultaneously tumor and blood BRCA NGS testing and, as soon as it is validated in the diagnostic setting, the HRD testing.

Even though this study presents some limitations, mainly due to its retrospective design, for our knowledge, it is one of the biggest real-world case series available. However, the blood test was not available for the whole study population, but only for selected patients, including patients with positive tumor BRCA testing or positive personal or family history and this may have introduced a selection bias. Someone may also point out the lack of follow up data and the unavailability of clinical data regarding the therapy response, which would be important to correlate the BRCA status with some clinical outcomes. This limit is in part due to the referral nature of many patients’ samples who were treated in other institutions.

## 5. Conclusions

Tumor BRCA testing is nowadays essential in the management of women affected by EOC. The implementation of a comprehensive and multidisciplinary clinical workflow is crucial to perform the test appropriately and timely. The proposed workflow may increase the capability to identify EOC patients with BRCA1/2 alteration maximizing the benefit of BRCA testing both from a therapeutic and risk assessment perspective. Since this field is continuously evolving, the workflow should be amenable to assimilate new advancements in technology, variant classification, patient stratification and drug availability. In the future, this will entail simultaneous tumor and blood BRCA NGS testing, and the evaluation of additional predictive biomarkers beyond BRCA1/2 gene alterations, as homologous recombination deficiency (HRD) signatures and immune biomarkers.

## Figures and Tables

**Figure 1 cancers-14-01638-f001:**
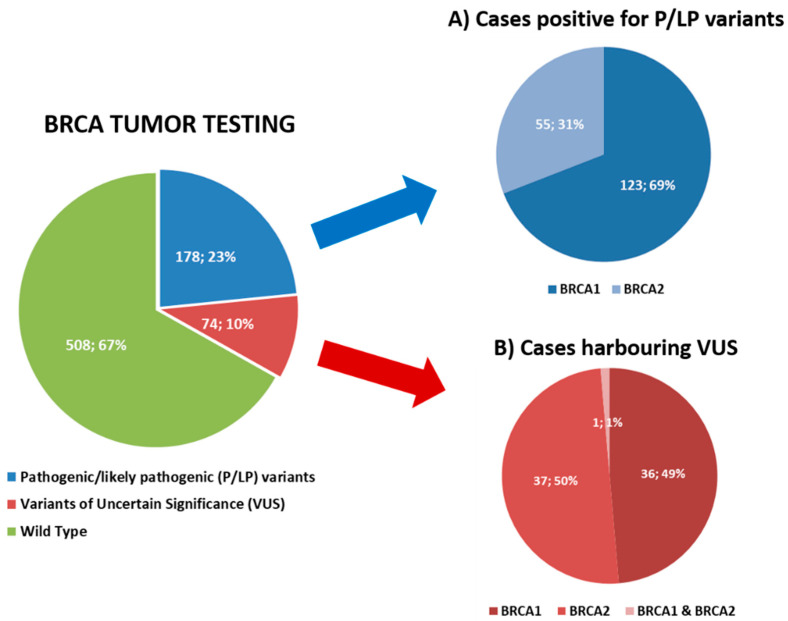
BRCA tumor testing results. Pie charts representing BRCA tumor testing results (number of cases; percentage). Distribution of cases harboring pathogenic/likely pathogenic variants (**A**) and variants of uncertain significance (**B**) affecting BRCA1 and BRCA2 genes.

**Figure 2 cancers-14-01638-f002:**
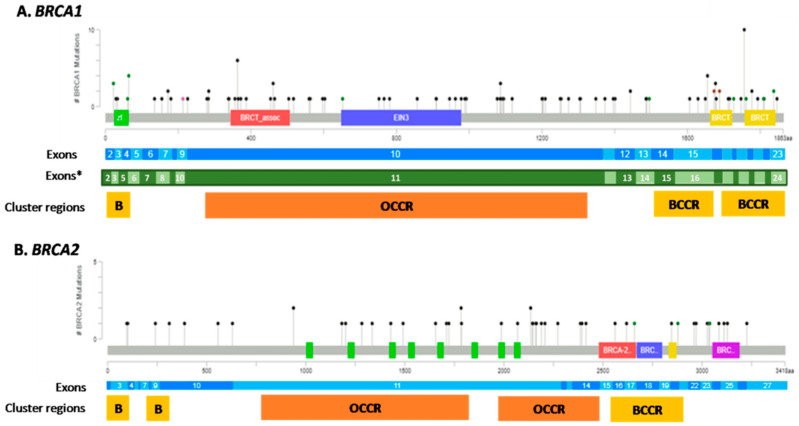
Lollipop plots of pathogenic/likely pathogenic variants mapped on BRCA1 (**A**) and BRCA2 (**B**) protein domains. BRCA1 domains: green, C3HC4 type RING finger; red, serine-rich domain associated with BRCT; blue, ethylene insensitive 3; yellow, BRCA1 C terminus domain. BRCA2 domains: green, BRCA2 repeats; red, BRCA2 helical; blue, BRCA2 oligonucleotide /oligosaccharide-binding domain 1; yellow, tower; purple, BRCA2 oligonucleotide/oligosaccharide-binding, domain 3. Each variant is represented by a single lollipop; the stick lengths indicate variant frequency (y-axis) and dots are color-coded according to variant type: green, missense mutations; black, truncating mutations (frameshift or nonsense mutations); brown dots, in-frame mutations. Intronic variants are not represented. Graphs created using Mutation Mapper tool, cBioportal (http://www.cbioportal.org/mutation_mapper) (accessed on 25 January 2022) and manually curated. BRCA1: RefSeq: NM_007294, Ensembl: ENST00000357654, CCDS: CCDS11453, UniProt: BRCA1_HUMAN. BRCA2:RefSeq: NM_000059, Ensembl: ENST00000380152, CCDS: CCDS9344, UniProt: BRCA2_HUMAN. * Exons number according traditional numbering, skipping exon 4 (1,2,3,5–24). Cluster regions: yellow BCCR: Breast cancer cluster region; orange OCCR: ovarian cancer cluster region.

**Figure 3 cancers-14-01638-f003:**
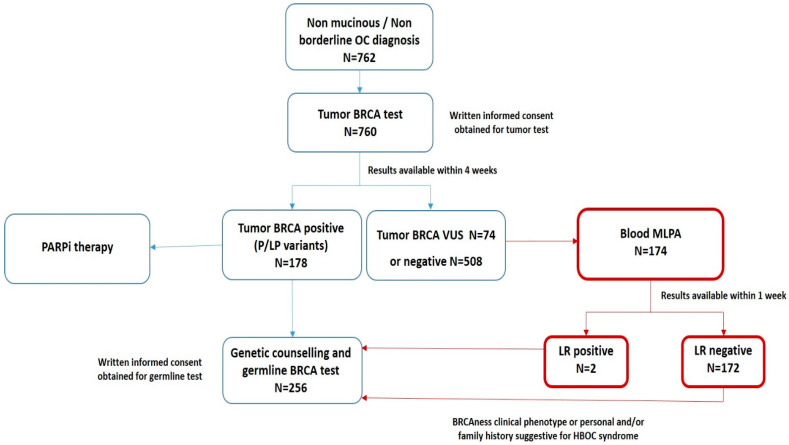
The new workflow for the management of women affected by ovarian cancer. Blue boxes: The workflow for the management of women affected by OC proposed in 2019 [20]. Red boxes: new steps introduced starting from January 2020. OC: ovarian cancer; P/LP: pathogenic/likely pathogenic variants; VUS: variant of uncertain significance; HBOC: hereditary breast and ovarian cancer syndrome; MLPA: Multiple-Ligation Probe-Amplification; LR: Large Rearrangements.

**Table 1 cancers-14-01638-t001:** Clinicopathological characteristics of the study cohort.

Clinicopathological Characteristics	Study Cohort (*n* = 762)
Age (Median, range)	61 (24–84)
Histological subtype	
High grade serous carcinoma	636 (83.5%)
Low grade serous carcinoma	17 (2.2%)
Clear cell carcinoma	30 (3.9%)
Endometrioid carcinoma	51 (6.7%)
Carcinoma—subtype not defined	28 (3.7%)
Figo stage	
I (A-C)	57 (7.5%)
II (A-B)	61 (8%)
III (A-C)	462 (60.6%)
IV (A-B)	151 (19.8%)
NA	31 (4.1%)
Tumor stage	
Early stage	82 (10.8%)
Advanced stage	649 (85.2%)
NA	31 (4.1%)
Family history	
Positive	427 (56%)
Negative	240 (31.5%)
NA	95 (12.5%)

**Table 2 cancers-14-01638-t002:** Characteristics of pathogenic/likely pathogenic variants and VUS. Molecular characteristics were tabulated overall and according to *BRCA1* or *BRCA2* genes affected.

	P/LP (*n* = 179 *) BRCA1 (*n* = 124 *), BRCA2 (*n* = 55)	VUS (*n* = 75 **) BRCA1 (*n* = 37 **), BRCA2 (*n* = 38 **)	*p*-Value
VAF			
(Median, range)	74% (5–99%)	47% (5–98%)	<0.0001 #
*BRCA1*	76.5% (6–99%)	44.5% (9–93%)	
*BRCA2*	63% (5–93%)	48% (5–98%)	
MUTATION TYPE			
(N, %)			<0.0001 #
Truncating	147 (82.1%)	4 (5.3%)	
*BRCA1*	101 (81.5%)	2 (5.4%)	
*BRCA2*	46 (83.6%)	2 (5.3%)	
Missense	18 (10.1%)	37 (49.3%)	
*BRCA1*	101 (81.5%)	13 (35.1%)	
*BRCA2*	46 (83.6%)	24 (63.2%)	
Splice site	7 (3.9%)	1 (1.3%)	
*BRCA1*	2 (1.6%)	1 (2.7%)	
*BRCA2*	5 (9.1%)	0	
Intronic	3 (1.7%)	28 (37.3%)	
*BRCA1*	2 (1.6%)	17 (45.9%)	
*BRCA2*	1 (1.8%)	11 (28.9%)	
Inframe	4 (2.2%)	5 (6.7%)	
*BRCA1*	4 (3.2%)	4 (10.8%)	
*BRCA2*	0	1 (2.6%)	

* 1 case with 2 BRCA1 pathogenic mutations. ** 1 case with 2 VUS alterations, affecting both *BRCA1* and *BRCA2*. # *p* value statistically significant.

**Table 3 cancers-14-01638-t003:** Clinicopathological features by BRCA status in the present cohort (N = 760). Number of cases (percentage row; percentage column). P/LP: pathogenic/likely pathogenic variants; VUS: variants of uncertain significance; WT: wild type; NA: Not Available.

	P/LP (*n* = 178)	VUS (*n* = 74)	WT (*n* = 508)	*p*-Value
Age	59 (37–81)	59 (39–79)	62 (24–84)	
(Median, range)				
Histological subtype				0.002 *
High grade serous carcinoma (*n* = 635)	168 (26.5%;944%)	60 (9.4%;81.1%)	407 (64.1%;80.1%)	
Low grade serous carcinoma (*n* = 17)	0	2 (11.8%;2.7%)	15 (88.2%;3%)	
Clear cell carcinoma (*n* = 30)	0	4 (13.3%;5.4%)	26 (86.7%;5.1%)	
Endometrioid carcinoma (*n* = 50)	6 (12%;3.4%)	2 (4%;2.7%)	42 (84%;8.3%)	
Carcinoma—subtype not defined (*n* = 28)	4 (14.3%; 2.2%)	6 (21.4%; 8.1%)	18 (64.3%;3.5%)	
Tumor status				0.79
Early (*n* = 81)	17 (21%; 9.6%)	8 (9.9%; 10.8%)	56 (69.1%; 11%)	
Advanced (*n* = 649)	152 (23.4%; 85.4%)	62 (9.6%; 83.8%)	435 (67%; 85.6%)	
NA (*n* = 30)	9 (30%; 5.1%)	4 (13.3%; 5.4%)	17 (56.6%; 3.3%)	
FIGO stage				0.11
I (A-C) (*n* = 56)	11 (19.6%; 6.2%)	3 (5.4%; 4.1%)	42 (75%; 8.3%)	
II (A-C) (*n* = 61)	7 (11.5%; 3.9%)	9 (14.8%; 12.2%)	45 (73.8%; 8.9%)	
III (A-C) (*n* = 462)	115 (24.9%; 64.6%)	38 (8.2%; 51.4%)	309 (66.9%; 60.8%)	
IV (A-B) (*n* = 151)	36 (23.8%; 20.2%)	20 (13.2%; 27%)	95 (62.9%; 18.7%)	
NA (*n* = 30)	9 (30%; 5.1%)	4 (13.3%; 5.4%)	17 (56.7%; 3.3%)	
Family history				0.007 *
Positive (*n* = 427)	118 (27.6%; 66.3%)	44 (10.3%; 59.5%)	265 (62.1%; 52.2%)	
Negative (*n* = 239)	49 (20.5%; 27.5%)	22 (9.2%; 29.7%)	168 (70.3%; 33.1%)	
NA (*n* = 94)	11 (11.7%; 6.2%)	8 (8.5%; 10.8%)	75 (79.8%; 14.8%)	

* *p*-value statistically significant.

**Table 4 cancers-14-01638-t004:** Crosstabulation of tumor and blood test results, available for 256 women affected by EOC. The number of cases and the percentages were tabulated according to the detection of pathogenic/likely pathogenic variants, VUS or negative BRCA status (wild type). B: blood test; T: tumor test; P/LP: pathogenic/likely pathogenic variants; VUS: Variants of Unknown clinical Significance; WT: Wild Type.

	B	P/LP (*n* = 100)	VUS (*n* = 6)	WT (*n* = 150)
T				
**P/LP (*n* = 132)**				
		100 (76%) #	-	32 (24%)

**VUS (*n* = 16)**		-	6 (37.5%) #	10 (62.5%)

**WT (*n* = 108)**		-	-	108 (100%) #


# Concordant tumor/blood BRCA status.

**Table 5 cancers-14-01638-t005:** Somatic and germline BRCA pathogenic/likely pathogenic variants. Distribution of clinicopathological features according to the somatic or germline nature of the P/LP alterations. Number of cases (percentage row; percentage column). P/LP: pathogenic/likely pathogenic variants; NA: Not Availabletest; P/LP: pathogenic/likely pathogenic variants; VUS: Variants of Unknown clinical Significance; WT: Wild Type.

	SOMATIC P/LP (*n* = 32; 24%)	GERMLINE P/LP (*n* = 100; 76%)	*p*-Value
Age (Median, range)	62.5 (40–79)	56 (37–78)	0.03 *
Histological subtype			
High grade serous carcinoma (*n* = 125)	28 (22.4%; 87.5%)	97 (77.6%; 97%)	0.04 *
Endometrioid carcinoma (*n* = 6)	4 (66.7%; 12.5%)	2 (33.3%; 2%)	
Carcinoma—subtype not defined (*n* = 1)	0	1 (100%; 1%)	
FIGO stage			
I (A-C) (*n* = 10)	4 (40%;12.5%)	6 (60%; 6%)	0.05 *
II (A-B) (*n* = 12)	7 (58.3%; 21.9%)	5 (41.7%; 5%)	
III (A-C) (*n* = 82)	12 (14.6%; 37.5%)	70 (85.4%; 70%)	
IV (A-B) (*n* = 27)	9 (33.3%; 28.1%)	18 (66.7%; 18%)	
NA (*n* = 1)	0	1 (100%; 1%)	
Tumor status			0.2
Early (*n* = 16)	6 (37.5%; 18.8%)	10 (62.5%; 10%)	
Advanced (*n* = 115)	26 (22.6%; 81.3%)	89 (77.4%; 89%)	
NA (*n* = 1)	0	1 (100%; 1%)	
Family history			
Positive (*n* = 95)	17 (17.9%; 53.1%)	78 (82.1%; 78%)	0.01 *
Negative (*n* = 34)	15 (44%; 46.9%)	19 (56%; 19%)	
NA (*n* = 3)	0 (0%)	3 (100%; 3%)	

* *p* value statistically significant.

## Data Availability

The data presented in this study are available in insert article and supplementary materials.

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
