# Peer review of "Tumor BRCA Testing in Epithelial Ovarian Cancers: Past and Future—Five-Years’ Single-Institution Experience of 762 Consecutive Patients"

_cancers, 2022, doi:10.3390/cancers14071638_

Round 1

Reviewer 1 Report

Interesting paper on BRCA tumor testing in a large serie of OC; precise analytical data on variants; providing a useful comprehensive worflow for clinical practice

minor comments : 

fig3 : why a arrow between negative LR and genetic counseling (particularly if no family history) ?

asking for written consent for tumor testing is a very relevant point (presented in figure 3) that could be emphasized in the discussion (not done everywhere)

table : for P value < 0.000001 or 10-6

Author Response

Reviewer 1

Interesting paper on BRCA tumor testing in a large serie of OC; precise analytical data on variants; providing a useful comprehensive worflow for clinical practice.

Minor comments :

  1. fig3 : why a arrow between negative LR and genetic counseling (particularly if no family history) ?

We thank the Reviewer for his comment. We suggested to further investigate patients with positive family or personal history suggestive for HBOC or with BRCAness clinical phenotype, defined as a phenotypic copy of BRCA mutations, which may underlie the presence of homologous recombination defects. These women can be addressed to genetic counselling for a wider evaluation.

  1. asking for written consent for tumor testing is a very relevant point (presented in figure 3) that could be emphasized in the discussion (not done everywhere)

We really thank the Reviewer for his kind suggestion. We add this crucial point in the discussion section. Please see page 11, lanes 352-354 “The test was suggested by specialists who explained the value and purposes of tumor testing to the patients and obtained a written informed consent.” and lanes 372-376: “During the genetic counselling, the geneticist may require a blood-based testing, describing to the patients the value of a blood test results in terms of risk assessment and following preventive actions for the woman itself and her relatives. For the blood-based BRCA testing a specific written consent may be obtained.”.

  1. table : for P value < 0.000001 or 10-6

We really thank the Reviewer for his kind suggestion. We change the p value in p < 0.0001, statistically significant, in Table 2.

Reviewer 2 Report

It is a single institution, but has a large number of cases. However, there are no new findings.

Author Response

It is a single institution, but has a large number of cases. However, there are no new findings

We thank the Reviewer for his comment. We reported our experience in tumor BRCA testing with a specific diagnostic algorithm based on a multidisciplinary approach.

Reviewer 3 Report

The authors present a follow-up study on their BRCA tumour testing experience, and an integrated workflow with germline genetics.

The study is well written, and sizable.  It is a nice study where MLPA is done in the blood due to the limitations of NGS in tumour.  A few minor comments.

  1. I would suggest a comprehensive review of the topic of BRCA tumour testing and the different yields in those studies and how the author's current study either confirms or builds on that knowledge.
  2. In the introduction, I would modify "one-drug fits all" to "drug and disease dyad."
  3. In figure 3, if the authors can put some quantities in the different cells of the flowsheet, that will give an idea of how the samples fell into different end categories.  MLPA should be specifically labelled as "blood MLPA."  Also, if some timelines on the different tests (e.g. MLPA, and genetic counselling/germline BRCA, would be helpful).
  4. For the MLPA rearrangement, did the authors look at the NGS data to see if those rearrangements could be detected?

Author Response

Reviewer 3

The authors present a follow-up study on their BRCA tumour testing experience, and an integrated workflow with germline genetics. The study is well written, and sizable.  It is a nice study where MLPA is done in the blood due to the limitations of NGS in tumour.  A few minor comments.

  1. I would suggest a comprehensive review of the topic of BRCA tumour testing and the different yields in those studies and how the author's current study either confirms or builds on that knowledge.

We thank the Reviewer for his comment. We insert a focus regarding this topic in the Discussion section. Please see page 10, lanes 287-297: “We confirmed the feasibility of tumor BRCA testing in the routine clinical setting, with a very low failure rate (less than 1%) and a turnaround time consistent with clinical need (median TAT less than 2 weeks), strengthening our previous results [21] and in line with the recently reported experience of other Italian groups [19,23,29]. However, both the studies of Marchetti [23] and Turchiano [29] were based on a fresh-frozen tissue approach, that was a very promising strategy to overcome the degradation and chemical modification of DNA resulting from formalin fixation and paraffin embedding, but that required framework and facilities (i.e. biobanking) not broadly available in hospitals. In the present study, according to the results presented by the Ligurian BRCA working group [19], we reported a workflow based on the analysis of FFPE specimens, available in every Pathology Division. In this context, the robustness of tumor BRCA testing is crucial.

  1. In the introduction, I would modify "one-drug fits all" to "drug and disease dyad."

We thank the Reviewer. We change the sentence according to his kind suggestion. Please see page 2, lanes 54-55.

  1. In figure 3, if the authors can put some quantities in the different cells of the flowsheet, that will give an idea of how the samples fell into different end categories.  MLPA should be specifically labelled as "blood MLPA."  Also, if some timelines on the different tests (e.g. MLPA, and genetic counselling/germline BRCA, would be helpful).

We really thank the Reviewer for his kind suggestions. We add the number of cases in each cell and timelines and change the MLPA in “blood MLPA”. Please see Figure 3_revised.

  1. For the MLPA rearrangement, did the authors look at the NGS data to see if those rearrangements could be detected?

We really thank the Reviewer for his question. We critically reviewed the Copy Number data obtained by NGS analysis of FFPE tumor samples of the patients with positive blood MLPA result. We confirmed that a copy number loss could be detected even in FFPE tumor samples.

Round 2

Reviewer 2 Report

The BRCA test in clinical practice is well described. It is a retrospective study, but I think it could be published.